# Does Total Neoadjuvant Therapy Impact Surgical Precision in Total Mesorectal Excision? A Nationwide Survey of the Experiences of Expert Surgeons

**DOI:** 10.3390/cancers17020283

**Published:** 2025-01-17

**Authors:** Tarkan Jäger, Matthias Zitt, Stefan Riss, Jaroslav Presl, Philipp Schredl, Daniel Neureiter, Jan Philipp Ramspott, Peter Tschann, Walter Brunner, Hermann Nehoda, Gerd Pressl, Klemens Rohregger, Robert Sucher, Gerhard Jenic, Andreas Heuberger, Reinhold Kafka-Ritsch, Jörg Tschmelitsch, Lukas Schabl, Isabella Dornauer, Florentina Dermuth, Karin Rokitte, Franz Singhartinger, Josef Holzinger, Ingmar Königsrainer, Klaus Emmanuel, Felix Aigner

**Affiliations:** 1Department of Surgery, Paracelsus Medical University Salzburg, 5020 Salzburg, Austria; j.presl@salk.at (J.P.); p.schredl@salk.at (P.S.); janphilipp.ramspott@gmail.com (J.P.R.); walter.brunner@kssg.ch (W.B.); l.schabl@salk.at (L.S.); i.dornauer@salk.at (I.D.); f.dermuth@salk.at (F.D.); k.rokitte@salk.at (K.R.); f.singhartinger@salk.at (F.S.); j.holzinger@salk.at (J.H.); k.emmanuel@salk.at (K.E.); 2Department of Surgery, Dornbirn General Hospital, 6850 Dornbirn, Austria; matthias.zitt@dornbirn.at; 3Department of General Surgery, Division of Visceral Surgery, Medical University Vienna, 1090 Vienna, Austria; stefan.riss@meduniwien.ac.at; 4Institute of Pathology, Paracelsus Medical University Salzburg, 5020 Salzburg, Austria; d.neureiter@salk.at; 5Department for General, Visceral and Transplant Surgery, University Hospital Muenster, 48149 Muenster, Germany; 6Department of General and Thoracic Surgery, Academic Teaching Hospital Feldkirch, 6800 Feldkirch, Austria; peter.tschann@lkhf.at (P.T.); ingmar.koenigsrainer@lkhf.at (I.K.); 7Department of General, Visceral, Endocrine and Transplant Surgery, Kantonsspital St. Gallen, 9007 St. Gallen, Switzerland; 8Department of General, Visceral and Vascular Surgery, Bezirkskrankenhaus St. Johann in Tirol, 6380 St. Johann in Tirol, Austria; nehoda@khsj.at; 9Department of Surgery, Ordensklinikum Linz, 4010 Linz, Austria; gerd.pressl@ordensklinikum.at (G.P.); klemens.rohregger@ordensklinikum.at (K.R.); 10Department of Surgery, Division of General, Visceral and Transplantation Surgery, Medical University of Graz, 8010 Graz, Austria; robert.sucher@medunigraz.at; 11Department of General and Vascular Surgery, Landeskrankenhaus Villach, 9500 Villach, Austria; gerhard.jenic@kabeg.at; 12Department of Surgery, Hospital Oberndorf, 5110 Oberndorf, Austria; a.heuberger@kh-oberndorf.at; 13Department of Visceral, Transplant and Thoracic Surgery, Medical University Innsbruck, 6020 Innsbruck, Austria; reinhold.kafka-ritsch@tirol-kliniken.at; 14Department of Surgery, Barmherzige Brüder Hospital St. Veit/Glan, 9300 St. Veit/Glan, Austria; joerg.tschmelitsch@bbstveit.at; 15Department of Surgery, Barmherzige Brüder Hospital Graz, 8020 Graz, Austria; felix.aigner@bbgraz.at

**Keywords:** TNT, total neoadjuvant therapy, neoadjuvant chemoradiotherapy, TME, total mesorectal excision, rectal cancer, surgical precision

## Abstract

In recent decades, the management of locally advanced rectal cancer has demonstrated significant advancements, resulting in improved patient outcomes. Total neoadjuvant therapy has gained increasing prominence, associated with enhanced progression-free survival and elevated rates of complete response. However, this intensified therapeutic approach introduces surgical complexities that remain incompletely elucidated. This study presents data derived from a nationwide survey of colorectal surgeons, focusing on surgical precision and outcomes related to total neoadjuvant therapy. The findings underscore the increased complexity of these procedures and emphasize the necessity for further research to optimize treatment strategies.

## 1. Introduction

Over the past four decades, the treatment of locally advanced rectal cancer (LARC) has evolved significantly, with continuous advancements aimed at improving the control of local and systemic disease (Figure 1) [1,2].

Initially, surgery combined with perioperative radiotherapy was the standard approach for treating LARC, resulting in high local recurrence rates of 30–40% [3,4]. The introduction of total mesorectal excision (TME) revolutionized rectal cancer treatment, reducing local recurrence rates to below 4%, thus emphasizing the critical importance of surgical precision [5]. Concurrently, preoperative radiotherapy further improved local control, as demonstrated in the Uppsala and Swedish Rectal Cancer trials [6,7]. However, it remained uncertain whether these improved outcomes were attributable to preoperative radiotherapy or TME, as TME had not yet been widely adopted.

The Dutch TME trial standardized the TME technique and demonstrated that combining preoperative short-course radiotherapy with TME reduced local recurrence rates to 2.4% [8]. Simultaneously, the German CAO/ARO/AIO-94 trial established the benefits of preoperative chemoradiotherapy, which reduced toxicity and improved local control compared to adjuvant therapy [9].

Over the past two decades, neoadjuvant long-course concurrent chemoradiation therapy or short-course radiation without chemotherapy, followed by TME and adjuvant chemotherapy have become the standard of care for LARC. While these approaches significantly improved local control, they failed to demonstrate substantial gains in long-term overall survival (OS).

To maximize the efficacy of adjuvant chemotherapy, the concept of total neoadjuvant therapy (TNT) was introduced, analogous to the rationale underlying the transition from post-operative to preoperative radiotherapy. TNT incorporates adjuvant chemotherapy into the neoadjuvant phase, facilitating the earlier treatment of micrometastatic disease and enhancing patient adherence. Recent studies have demonstrated that TNT improves progression-free survival and complete response rates, although its efficacy remains contingent upon patient selection [10,11,12,13].

Based on four key premises, we have conducted a nationwide survey among colorectal cancer surgeons to evaluate the impact of intensified neoadjuvant therapy under the TNT approach, particularly focusing on its perceived effects on surgical precision, especially the quality of TME, and post-operative morbidity.

Neoadjuvant Therapy for All Patients: While neoadjuvant therapy can improve locoregional control, it is not universally necessary for all patients. For instance, in low-risk patients with cT3 tumors and limited extramural spread, excellent outcomes can be achieved with TME alone, thereby avoiding potential treatment-related complications [14]. This suggests that TNT, especially in low-risk cases, may result in overtreatment.Controversy Surrounding Adjuvant Chemotherapy: The role of adjuvant chemotherapy in the treatment of LARC remains contentious with wide variation in clinical practice. Although three-quarters of patients receive adjuvant chemotherapy after TME, fewer than half complete the planned treatment [15]. Differences in patient selection criteria and chemotherapy regimens contribute to concerns about overtreatment, particularly when chemotherapy is administered as part of a neoadjuvant strategy, such as in the TNT concept.Surgical Precision and TME: The success of Heald’s pioneering work in achieving outstanding surgical outcomes without the need for radiotherapy highlights the critical importance of surgical precision in rectal cancer treatment [16]. TME remains a cornerstone procedure in reducing local recurrence and improving patient outcomes, emphasizing that high-quality surgery is paramount even in the context of evolving multimodal treatments [17]. Thus, any intervention that potentially interferes with surgical precision should be implemented with careful consideration.Long-term Follow-up and Recurrence: Comprehensive long-term data on local recurrence after TNT remain limited, particularly as local recurrence has been observed to occur more than 8 years after conventional neoadjuvant therapy (CNT) followed by TME surgery [18]. The elevated local recurrence rates observed in the 5-year follow-up of the RAPIDO trial in the experimental arm (10% vs. 6%, *p* = 0.027), despite achieving a clinical complete remission rate of 28%, raise concerns regarding the potential impact of intensified neoadjuvant therapy on surgical quality [19]. It is noteworthy that these results were not available or published at the time the survey was conceived.

We postulate that the intensified treatment regimen, in conjunction with the prolonged time interval (>22 weeks), may adversely affect the precision of TME. This effect is particularly evident in reduced visibility and the disruption of correct tissue plane identification, potentially leading to a higher rate of suboptimal Mercury classification, increased R1 resections, and a greater risk of tumor perforation. In this context, the quality of surgery, particularly TME, becomes increasingly critical in controlling local recurrence. A high-quality TME is essential to mitigate the risks introduced by intensified neoadjuvant therapies and to preserve long-term oncological outcomes.

By focusing on these aspects, the survey aims to gather insights from practicing colorectal surgeons regarding the perceived challenges and outcomes associated with TNT, specifically in terms of maintaining surgical quality and managing post-operative complications.

## 2. Materials and Methods

This survey was designed to obtain comprehensive insights into the experiences and practices related to the treatment of LARC utilizing both CNT and TNT approaches. The survey comprised various sections, each aimed at investigating a specific aspect of LARC treatment, from surgical techniques to post-operative outcomes.

Data were collected through an online questionnaire utilizing the formr framework, which facilitated efficient survey distribution and real-time response collection [20]. The questionnaire incorporated multiple-choice questions, Likert scale items, and open-ended questions to gather both quantitative and qualitative data.

A panel of five core members selected 50 oncologic surgeons from the Austrian Society of Surgical Oncology (ACO-ASSO) colorectal branch member list, actively engaged in rectal cancer surgery for the initial survey distribution. To enhance the representativeness of the sample, the list was subsequently expanded to include 57 participants. This selection process aimed to provide a comprehensive cross-section of experts in the field of colorectal surgery. The survey was initiated on 31 May 2024, and a total of 57 surgeons were invited to participate. The survey remained open for four weeks with one automated reminder email sent seven days after the initial invitation.

The survey was structured into five main categories, each addressing different aspects of LARC treatment. The first category comprised seven general questions about the overall management and multidisciplinary approaches to rectal cancer treatment. The second category, containing six questions, aimed to collect data on surgical outcomes and protocols following CNT, comparing short-course radiation followed by TME with long-course chemoradiotherapy followed by TME. The third category included eight questions focusing on the influence of TNT on surgical practices and patient outcomes. The fourth and fifth categories employed a 5-point Likert scale to assess surgeons’ perceptions of surgical challenges. The fourth category addressed the management of patients who did not achieve complete clinical remission following TNT, while the fifth category concentrated on managing regrowth in patients following a watch-and-wait strategy post-TNT.

To investigate potential disparities in the perception of surgical challenges, we conducted a subgroup analysis comparing high-volume centers (managing >40 primary rectal cancer cases annually) and low-volume centers (managing ≤40 cases annually). Responses for Scenario 1 (TNT without complete clinical remission) and Scenario 2 (regrowth under watch-and-wait) were analyzed for disagreement rates (sum of ‘Strongly Disagree’ and ‘Disagree’). The results were compared between the two groups, and overall trends were interpreted.

All statistical analyses and graphical outputs were performed using R version 4.3.1 [21]. The detailed questionnaire utilized in this study is available as a Appendix A. Participation was voluntary, and all responses were anonymized to ensure confidentiality. The survey adhered to the ethical guidelines of the ACO-ASSO.

## 3. Results

A total of 31 responses were received, resulting in an overall response rate of 54.4%. Of the 31 respondents, 30 provided eligible responses regarding experiences with CNT, 28 for TNT, and 17 for surgery after regrowth.

All 31 respondents routinely present patients with primary rectal cancer at a multidisciplinary tumor board in their respective hospitals. The plurality of respondents (35%) reported 21–40 primary rectal cancer cases presented annually (Figure 2). Forty-two percent of hospitals were classified as high-volume centers, performing more than 40 primary rectal cancer cases annually, while 58% were classified as low-volume centers, handling 40 or fewer cases per year. Additionally, 81% of respondents confirmed the routine implementation of post-operative histopathological MERCURY assessment in their hospitals.

CNT was the standard of care for the majority of respondents (56%), with 32% utilizing chemoradiotherapy and 24% employing short-term radiotherapy. The two most preferred therapeutic strategies in the neoadjuvant or global setting were chemoradiotherapy and TNT, selected by 32% and 31% of respondents, respectively (Figure 3).

The majority of surgical procedures following CNT (59%) were conducted 8–10 weeks post-CNT. Conversely, 31% of respondents reported initiating surgery earlier, between 6–7 weeks after TNT. An additional 28% scheduled surgical interventions 8–10 weeks post-TNT. Beyond these time frames, the timing of surgical procedures varied among respondents in both cohorts (Figure 4). Regarding the preferred sequencing of neoadjuvant components in the TNT approach, 47% (14/30) of respondents opted for a variable strategy, 33% favored consolidation (radiation first), while 20% selected induction (chemotherapy first).

The majority of respondents employed robot-assisted surgery, with 50% utilizing it post-CNT and 50% post-TNT. For detailed proportions of alternative surgical approaches, please refer to Figure 5. These findings suggest a notable trend towards robot-assisted surgery, regardless of the neoadjuvant approach employed (Figure 5).

A consistent high rate of diverting stoma creation was observed post-CNT (77%) and post-TNT (82%) (Figure 6). The preferred stoma type was ileostomy in both groups, with 87% post-CNT and 86% post-TNT, compared to transversostomy at 13% and 14%, respectively. The creation of a stoma was primarily influenced by tumor height (50% post-CNT, 54% post-TNT), previous therapy (37% post-CNT, 35% post-TNT), and other factors (13% post-CNT, 11% post-TNT).

### 3.1. Scenario 1—Surgery After TNT Without Complete Clinical Remission

For the scenario involving TME after TNT without a complete clinical remission, the respondents’ experiences and perceptions were as follows:

Regarding general experiences, 65% of respondents (29% strongly) reported challenges with tissue dissection. Conversely, 18% remained neutral, and another 18% (4% strongly) observed no differences. Furthermore, 57% of respondents (14% strongly) encountered difficulties with the identification of tissue planes, whereas 25% reported no difficulties (7% strongly). Approximately one-third of respondents reported increased bleeding during surgery (7% strongly). Tissue fragility was identified as a concern by 47% (11% strongly) of respondents. Conversely, 25% remained neutral, and 29% (4% strongly) disagreed (Figure 7).

Of the respondents, 39% (21% strongly) reported that TME quality after TNT differed compared to TME after CNT. Conversely, 32% (14% strongly) indicated that the quality was similar, and 29% remained neutral. These findings suggest that a greater proportion of respondents perceived differences in TME quality post-TNT compared to post-CNT. Regarding the question of whether TME quality post-TNT was superior to post-CNT, 67% (46% strongly) disagreed. Additionally, 29% of respondents were neutral, and only 4% agreed. When assessing whether the quality of TME post-TNT was inferior to post-CNT, 35% (14% strongly) agreed, 28% (21% strongly) disagreed, and 36% remained neutral (Figure 8).

Regarding anastomotic leakage after TNT, 50% (25% strongly) concurred that no abnormalities were observed, while 22% (11% strongly) disagreed, and 29% remained neutral. Specifically, 53% (32% strongly) of respondents disagreed that there was an increased risk of anastomotic leakage, with 29% neutral, and only 18% (4% strongly) in agreement. The perception of increased anastomotic leakage after TNT was not widely held (Figure 9).

A significant proportion of respondents (35%, with 14% strongly agreeing) reported increased wound healing complications following abdominoperineal excision, while 32% (21% strongly) disagreed. The remaining 32% expressed a neutral stance, indicating a diverse range of experiences regarding wound healing issues (Figure 10).

### 3.2. Scenario 2—Surgery After Regrowth Following Watch-and-Wait After TNT

The survey also examined surgical outcomes following regrowth after a watch-and-wait strategy for patients with an initially clinically complete response to TNT.

A significant 64% of respondents disagreed (29% strongly) that there was no quality difference in tissue dissection during TME after TNT and regrowth, indicating that the majority of respondents observed quality changes in tissue dissection, consistent with Scenario 1.

Difficulties identifying surgical planes were reported by 59% of respondents (18% strongly). Conversely, 30% did not encounter difficulties, similar to Scenario 1, where a majority experienced challenges with plane identification. Increased bleeding during surgery was not a primary concern, as 41% of respondents did not encounter bleeding issues, while 30% experienced bleeding issues (12% strongly).

Consistent with Scenario 1, 47% of respondents (18% strongly) reported higher tissue fragility. The perception of worsening TME quality was similar to Scenario 1, as indicated by 36% of respondents. Increased anastomotic leakage and wound healing issues were reported by 24% and 42%, respectively, both slightly higher than in Scenario 1 (Figure 11).

Overall, the results of Scenario 2 were largely consistent with those of Scenario 1, suggesting similar experiences and perceptions among respondents, with only minor variations.

To address potential differences in the perception of surgical challenges between high-volume and low-volume centers, a subgroup analysis was conducted for Scenario 1 and Scenario 2. The results are summarized in Table 1. While no statistically significant differences were observed, trends emerged that reflect both group-specific responses and overall perceptions.

In Scenario 1, disagreement rates revealed that surgeons in high-volume centers were less likely to report “difficulties in surgical plane identification” (31% vs. 20%) and “fragile tissue, tears easily” (38% vs. 20%) compared to low-volume centers. However, disagreement with “Yes, better quality than CNT” was higher in high-volume centers (77%) than in low-volume centers (60%), suggesting greater skepticism regarding the improvement of TME quality following TNT. For the item “Yes, worse quality than CNT”, only 46% in high-volume centers disagreed, compared to 13% in low-volume centers, indicating that a larger proportion of surgeons in low-volume centers perceived TNT as negatively impacting surgical quality (Figure 12).

In Scenario 2, fewer disparities were observed overall. For instance, disagreement with “increased bleeding” was more prevalent in high-volume centers (56%) compared to low-volume centers (25%). Similarly, disagreement with “anastomotic insufficiencies” was noted in 56% of high-volume centers and 38% of low-volume centers. Despite minor variations, both groups acknowledged challenges such as bleeding and tissue-related issues while maintaining overall reservations regarding the quality of TME following TNT compared to CNT (Figure 13).

## 4. Discussion

We conducted a nationwide survey among colorectal cancer surgeons to evaluate the impact of intensified neoadjuvant therapy under the TNT approach, focusing on its perceived effects on surgical precision, particularly the quality of TME, and post-operative morbidity.

In our survey, a significant proportion of respondents reported difficulties with tissue dissection (65%) and the identification of surgical planes (57%), suggesting that TNT may fundamentally alter tissue characteristics, leading to complications and decreased precision during TME. Concerns regarding tissue fragility (47%) indicate that TNT-treated tissues may require distinct handling techniques compared to CNT-treated or untreated tissues. Furthermore, approximately one-third of respondents noted increased bleeding tendency, suggesting an inconsistent yet significant impact on tissue vascularity, further reducing surgical precision during TME. These findings were observed in both scenarios (scenario 1: TNT without complete clinical remission, and scenario 2: regrowth during the watch-and-wait strategy post-TNT).

It is crucial to emphasize that any preoperative therapy can affect the quality of the subsequent TME, which in turn influences local recurrence rates and ultimately overall patient outcomes [17].

Even prior to the TNT era, CNT approaches, such as long-course concurrent chemoradiation or short-course radiation without chemotherapy, presented significant challenges, often compromising surgical precision. These challenges primarily arose due to increased tissue fragility and fibrosis, which complicated dissection and surgical accuracy [22]. Furthermore, extended intervals between radiotherapy and TME have been associated with higher rates of fibrosis, further compromising surgical precision. The Timing of Rectal Cancer Response to Chemoradiation Consortium reported increased pelvic fibrosis and surgical challenges with longer radiation–surgery intervals [23]. Similarly, the GRECCAR-6 trial demonstrated increased morbidity and technical difficulties with an eleven-week interval compared to a seven-week interval [22]. The CRONOS study identified three key time intervals—short (≤8 weeks), intermediate (>8 to ≤12 weeks), and long (>12 weeks)—each associated with different outcomes. While longer intervals improved tumor regression grades and reduced systemic recurrence, they increased surgical complexity and minor morbidity [24].

In contrast to the pre-TNT era, where CNT was followed by TME and subsequent adjuvant chemotherapy with uniform time intervals, the TNT concept repositions systemic chemotherapy ahead of the TME phase, either through induction (chemotherapy before radiation) or consolidation (radiation before chemotherapy). Depending on the study protocol, this approach may be followed by TME or a watch-and-wait strategy, resulting in variable intervals and conditions that can impact surgical precision [10,12].

The timing for CNT is typically well-defined, with surgery generally performed 6–8 weeks after long-term chemoradiotherapy or immediately following short-course radiotherapy. However, the timing after TNT exhibits greater variability, primarily contingent upon the specific protocol employed. For instance, in the RAPIDO trial, which utilized consolidation chemotherapy, surgical intervention was scheduled 2–4 weeks subsequent to the completion of consolidation chemotherapy [10]. Similarly, in the STELLAR trial, which also incorporated consolidation chemotherapy, TME surgery was recommended 6–8 weeks following preoperative treatment [13]. In contrast, the PRODIGE 23 trial, which employed induction chemotherapy, stipulated mandatory surgery 6–8 weeks after chemoradiotherapy. Notably, with the exception of the STELLAR trial—where a “watch and wait” strategy was permissible in cases of clinical complete response—all of these trials mandated TME surgery [13]. A distinct approach was adopted in the OPRA trial, which encompassed both induction and consolidation chemotherapy arms [12]. This trial recommended TME surgery for patients exhibiting an incomplete clinical response at tumor restaging 8 weeks (±4 weeks) subsequent to the completion of TNT.

The optimal regimen remains undetermined, as direct comparisons between different TNT regimens are limited. The selection of the most appropriate approach is yet to be definitively established and necessitates multidisciplinary decision-making based on both local and systemic risk factors, ensuring that therapy is tailored to individual patients and their tumor characteristics [25,26,27].

In our survey, 47% reported a variable approach to selecting consolidation or induction chemotherapy, while 33% preferred consolidation and 20% favored induction chemotherapy. These findings highlight significant variability in clinical practice regarding the utilization of consolidation and induction chemotherapy in TNT, which subsequently results in differences in the timing of surgery after radiotherapy. For instance, in the case of consolidation chemotherapy, as employed in RAPIDO [10] and STELLAR [13], extended total intervals between radiotherapy and surgery are introduced due to the additional duration of chemotherapy (e.g., 18 weeks in RAPIDO [10]). These findings underscore that both the interval itself and the selected therapeutic strategy play crucial roles in determining the timing and potentially the outcomes of surgery.

Both the additional chemotherapy administered before surgery as part of the TNT protocol and the prolonged intervals before surgery present significant risks to surgical precision and patient outcomes.

The subgroup analysis comparing high-volume and low-volume centers provides additional insight into the perception of surgical challenges following TNT. Although no statistically significant differences were found, the observed trends elucidate nuanced differences based on institutional experience. High-volume centers reported higher disagreement with “better quality than CNT” and lower disagreement with “worse quality than CNT”, suggesting a greater sensitivity to subtle declines in TME quality. This phenomenon may be attributed to their experience in managing more complex cases or heightened awareness of surgical precision.

In contrast, low-volume centers demonstrated a higher likelihood of perceiving TNT as having a negative impact on TME quality, as evidenced by the lower disagreement rates with “worse quality than CNT”. Additionally, challenges such as tissue fragility and wound healing issues appeared more prevalent in low-volume centers.

In Scenario 2, which addressed regrowth following a watch-and-wait strategy, differences between the groups were less pronounced, with “No Difference” being the most frequent response. However, challenges such as bleeding and anastomotic insufficiencies were still reported more frequently in high-volume centers, potentially reflecting case complexity or enhanced complication detection.

These findings underscore the shared recognition of challenges associated with TNT, while highlighting the role of institutional experience in shaping perceptions. The subgroup analysis provides valuable context to the variability in responses and emphasizes the need for further studies to assess surgical outcomes more objectively.

The disparity between disagreement with “better TME quality after TNT” and agreement with “worse TME quality after TNT” suggests that while the majority of surgeons do not perceive an improvement, over one-third reported a decline, which is clinically significant. Subgroup analysis revealed that high-volume centers showed greater sensitivity to subtle declines, while low-volume centers more frequently reported worsened quality, likely reflecting differences in surgical complexity and experience. Furthermore, the timing of surgery—particularly prolonged intervals in consolidation chemotherapy protocols—may exacerbate tissue fibrosis and complicate TME, highlighting the need for optimized timing strategies and standardized protocols to minimize these challenges.

Xu et al. conducted a retrospective cohort study comparing the effects of TNT versus neoadjuvant chemoradiation on post-operative outcomes in stage II/III rectal cancer patients. The study, involving 181 patients, found that TNT was associated with longer operative times and increased estimated blood loss; however, there were no significant differences in severe post-operative complications, complete TME rates, or negative circumferential margin (CRM) rates compared to neoadjuvant chemoradiation. Despite the extended operative duration and higher blood loss linked to TNT, these differences were minor and should be interpreted cautiously [28].

A systematic review and meta-analysis by Kasi et al. compared TNT with standard chemoradiotherapy followed by surgery and adjuvant chemotherapy in patients with LARC. TNT was associated with a higher rate of pathologic complete response (29.9% vs. 14.9%) and improved disease-free survival but demonstrated no significant differences in sphincter-preserving surgery or ileostomy rates compared to the standard arm. The long-term effects on OS and disease recurrence remain uncertain and necessitate further investigation [29].

Similarly, Goffredo et al. evaluated the impact of total TNT versus neoadjuvant chemoradiation followed by adjuvant chemotherapy on OS, tumor downstaging, and CRM status in 8548 LARC patients. TNT resulted in higher rates of tumor downstaging but did not improve OS or CRM status compared to CNT [30].

Despite high pathological complete response rates, the RAPIDO trial demonstrated significantly higher local recurrence rates at the 5-year follow-up in the TNT arm compared to CNT (10% vs. 6%, *p* = 0.027) [19]. This finding aligns with the observed alterations during TME after TNT, where theoretically different time intervals before surgery may contribute to two distinct tissue states: initially fragile tissue with increased bleeding tendency immediately after intensified TNT, followed by fibrosis and a loss of clear surgical planes over time. These impairments may contribute to worse long-term outcomes, such as increased local recurrence rates, reinforcing the established critical role of TME quality in controlling recurrence [5,17].

While TNT aims to enhance systemic control and reduce metastasis, it appears to affect the surgical quality of TME, as indicated by the MERCURY classification. The findings of our survey underscore the perception that TNT may lead to diminished TME quality compared to CNT. Specifically, 68% of respondents disagreed that TNT resulted in superior TME quality than CNT, while 36% agreed that TME quality was inferior, which could be attributed to factors such as tissue fragility, fibrosis, and altered tissue handling. This concern is particularly pertinent when considering intensified neoadjuvant therapy for locally non-advanced tumors, especially for organ preservation in T1 and T2 rectal cancers.

These insights underscore the necessity for ongoing evaluation and the potential development of novel surgical techniques or protocols to address the unique challenges introduced by TNT, specifically those related to enhanced tissue alterations from intensive pre-surgical treatment strategies.

However, several limitations must be considered regarding the methodology of the survey. The results are based on subjective responses from colorectal surgeons, which may introduce bias, particularly due to individual experiences and perspectives. Additionally, the sample may not fully represent all practitioners involved in colorectal cancer treatment, limiting the generalizability of the findings. Furthermore, the survey relied on retrospective perceptions rather than objective measurements, making it challenging to quantify the precise impact of TNT on surgical quality. Another limitation is that the survey did not distinguish between consolidation and induction chemotherapy within the TNT approach, which may have led to variability in responses and limited the ability to draw precise conclusions about their specific effects on timing and outcomes.

Additionally, variability in tissue changes may arise not only from differences in therapy modalities—such as the type, dosage, or duration of neoadjuvant therapy—but also from patient-specific factors, including baseline health, comorbidities, and genetic predispositions. These factors were not accounted for in this survey and may independently influence tissue response and surgical outcomes. The survey did not include questions regarding conversion rates, which may have provided additional insights into the practical impact of TNT, particularly in cases where difficulties such as surgical plane identification are encountered.

## 5. Conclusions

Our findings suggest that TNT in the treatment of LARC presents considerable perceived technical challenges during TME, particularly with respect to tissue fragility, difficulties in surgical plane identification, and overall tissue handling. This observation contradicts the established understanding that surgical quality is a critical determinant in the local recurrence rate of LARC treatment, emphasizing that, notwithstanding the systemic benefits of TNT, maintaining surgical precision remains essential to minimize the risk of local recurrence. Our observations are further corroborated by the long-term results of the RAPIDO trial, which demonstrated significantly increased local recurrence rates in the experimental TNT arm [19]. The concurrent shift from CNT to TNT, in conjunction with the transition from laparoscopic to robotic surgery, may introduce unaccounted biases in the evaluation of local recurrence rates. Addressing these challenges necessitates refined protocols and further research to optimize treatment strategies and improve outcomes in patients with LARC.

## Figures and Tables

**Figure 1 cancers-17-00283-f001:**
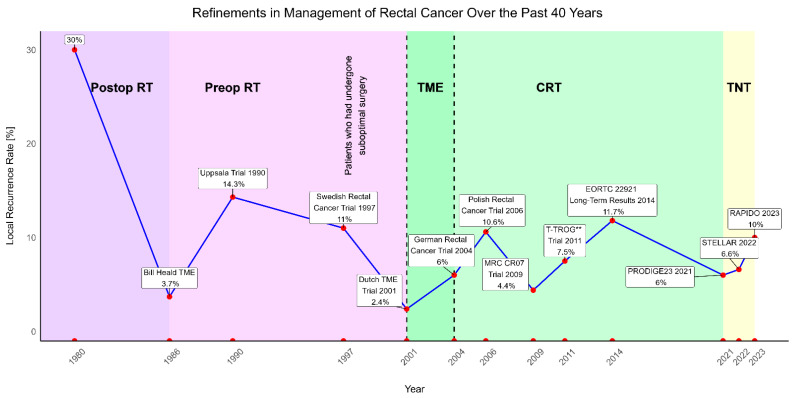
Phases of development in locally advanced rectal cancer treatment over the past 40 years. Key studies and local recurrence rates (y-axis) are shown. CRT = Chemoradiotherapy, EORTC = European Organisation for Research and Treatment of Cancer, MRC = Medical Research Council, RT = Radiotherapy, TME = total mesorectal excision, TNT = total neoadjuvant therapy, and ** = Trans-Tasman Radiation Oncology Group.

**Figure 2 cancers-17-00283-f002:**
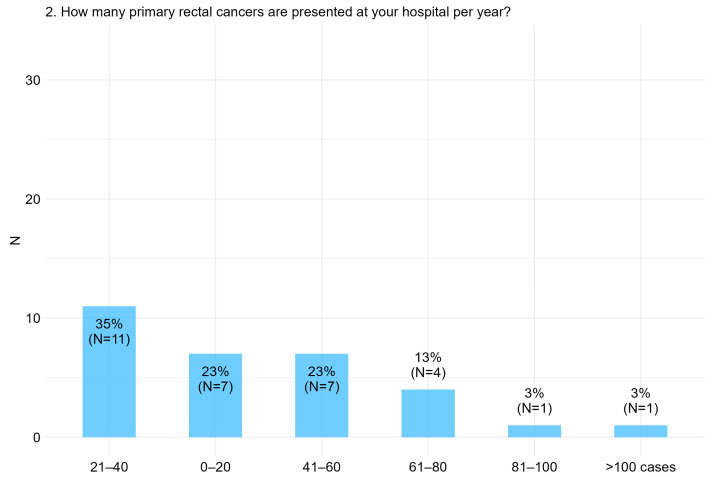
Item I.2: How many primary rectal cancer cases are presented in your hospital per year?

**Figure 3 cancers-17-00283-f003:**
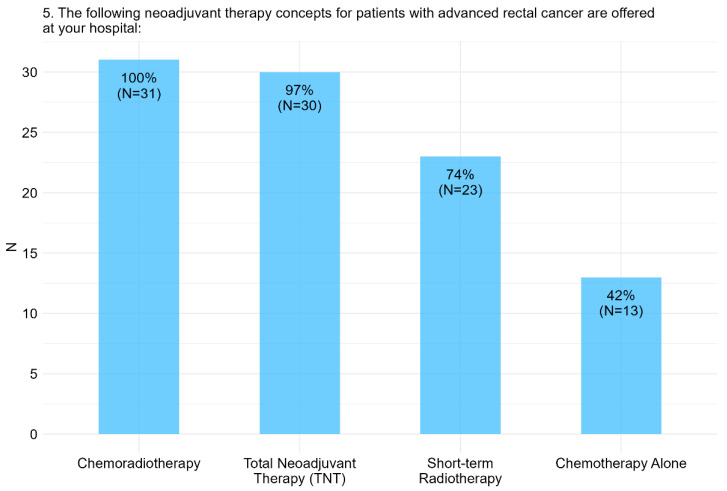
Item I.5: The following neoadjuvant therapy concepts for patients with advanced rectal cancer are offered at surveyed hospitals. Percentages are calculated relative to the total number of participants (*n* = 31), reflecting the proportion of respondents selecting each option in a multiple-choice format.

**Figure 4 cancers-17-00283-f004:**
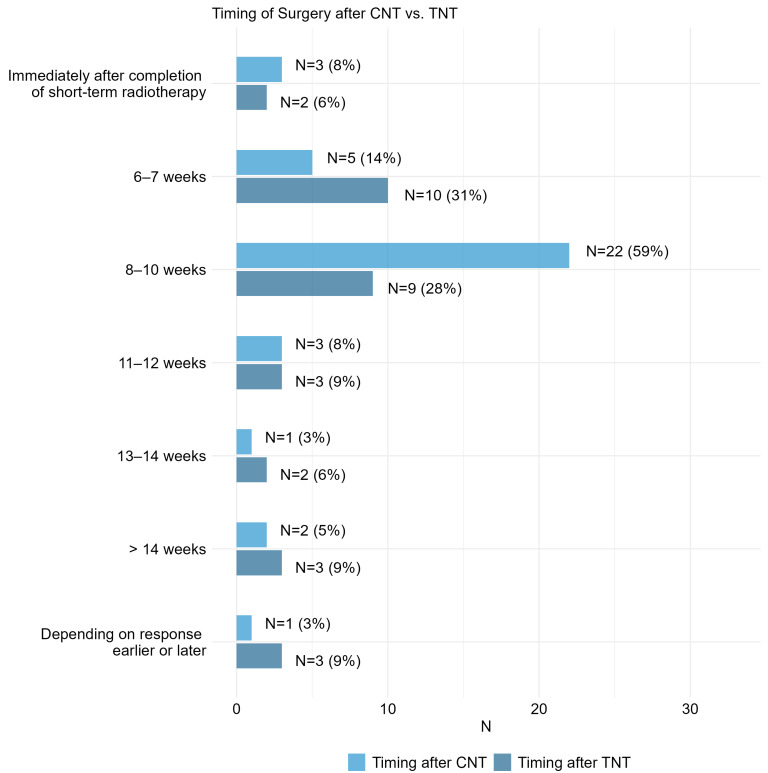
Item II.2 and III.2: When do you generally operate on patients with residual tumors after CNT vs. TNT? Multiple answers possible. CNT = conventional neoadjuvant therapy, TNT = total neoadjuvant therapy.

**Figure 5 cancers-17-00283-f005:**
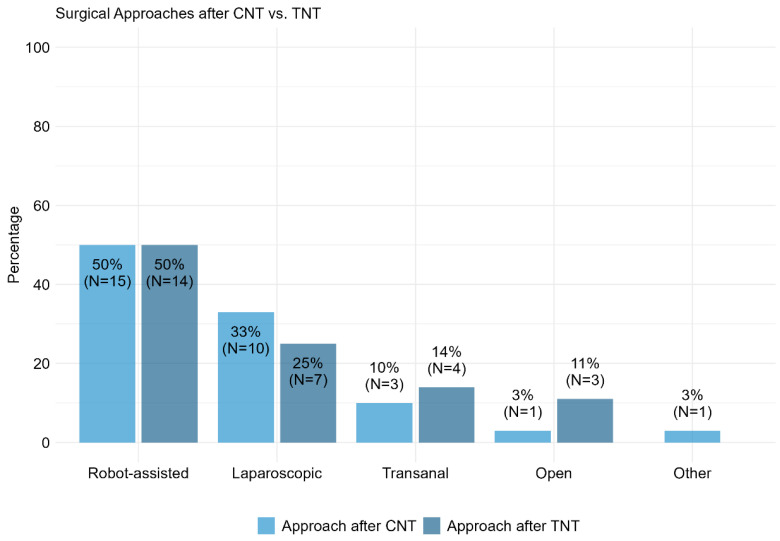
Item II.3 and III.3: What surgical technique did you prefer for your most recent CNT vs. TNT patients? CNT = conventional neoadjuvant therapy, TNT = total neoadjuvant therapy.

**Figure 6 cancers-17-00283-f006:**
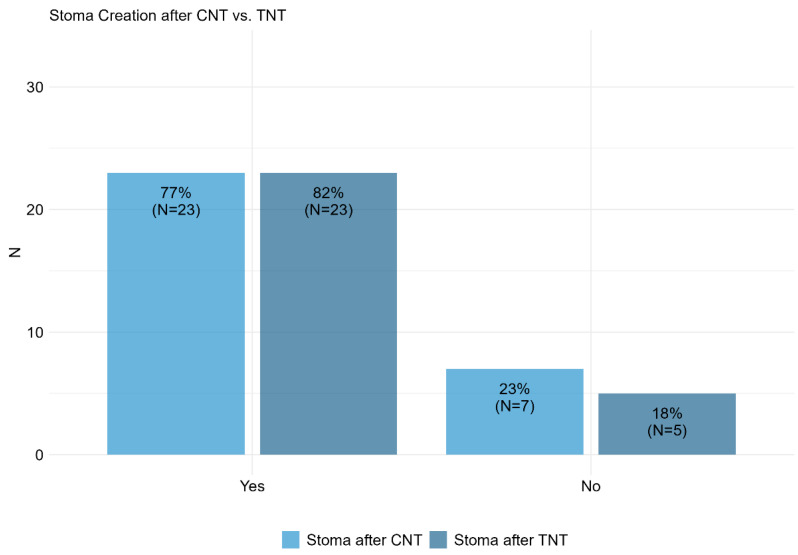
Item II.4 and III.4: Do you routinely create a protective stoma after CNT vs. TNT? CNT = conventional neoadjuvant therapy, TNT = total neoadjuvant therapy.

**Figure 7 cancers-17-00283-f007:**
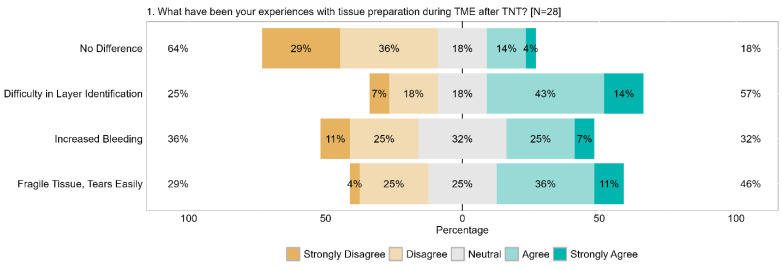
Item IV.1: What experiences have you had during TME after TNT? TME = total mesorectal excision, TNT = total neoadjuvant therapy.

**Figure 8 cancers-17-00283-f008:**
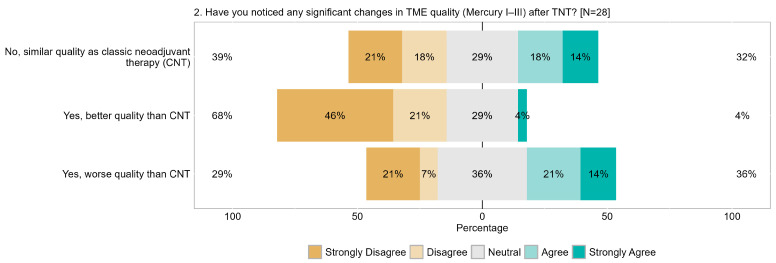
Item IV.2: Have you noticed any significant changes in TME quality (according to Mercury I–III) after TNT? TME = total mesorectal excision, TNT = total neoadjuvant therapy.

**Figure 9 cancers-17-00283-f009:**
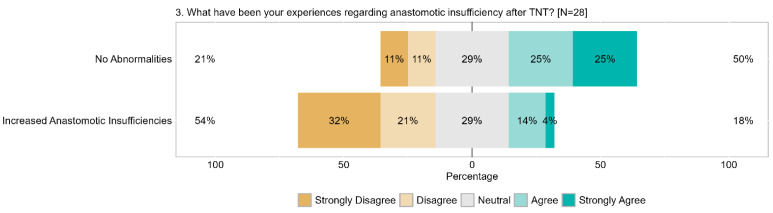
Item IV.3: What experiences have you had regarding anastomotic insufficiency after TNT? TNT = total neoadjuvant therapy.

**Figure 10 cancers-17-00283-f010:**
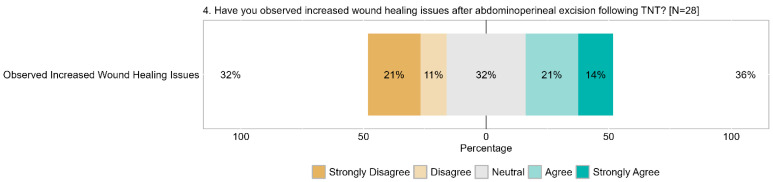
Item IV.4: Have you observed increased wound healing issues after abdominoperineal excision following TNT? TNT = total neoadjuvant therapy.

**Figure 11 cancers-17-00283-f011:**
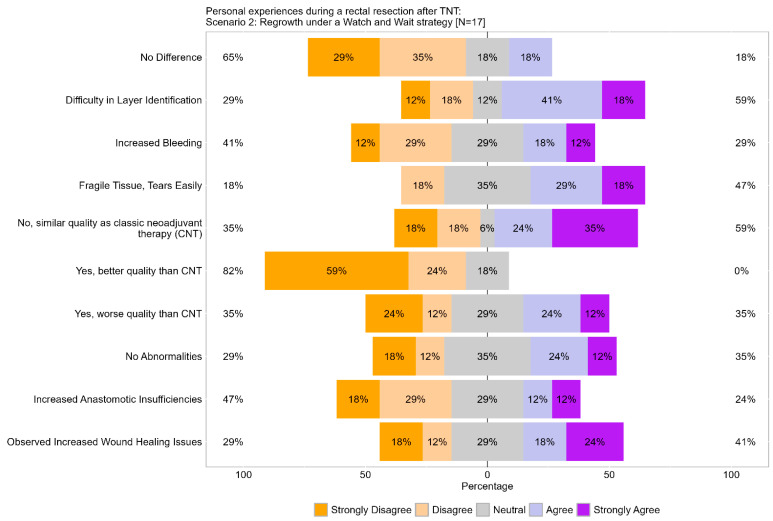
Item V.1–V.4: Personal experiences during a rectal resection after TNT in the regrowth scenario under watch-and-wait strategy. TNT = total neoadjuvant therapy.

**Figure 12 cancers-17-00283-f012:**
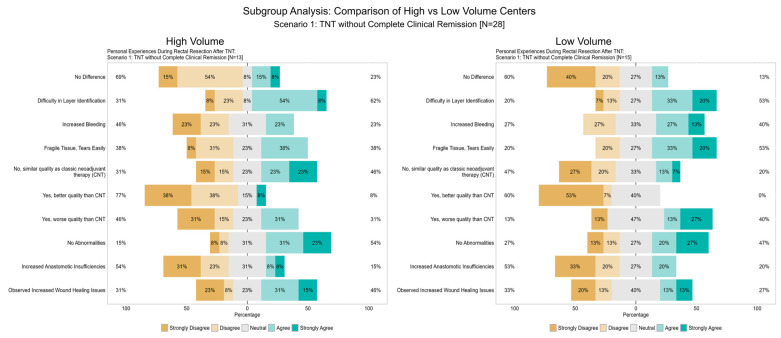
Item IV.1–IV.4: Subgroup analysis of personal experiences during a rectal resection after total neoadjuvant therapy (TNT) (Scenario 1). The figure compares responses from surgeons in high-volume (**left**) and low-volume (**right**) centers regarding their personal experiences during rectal resection after TNT without complete clinical remission (Scenario 1, N = 28). Responses are presented as percentages across five Likert scale categories: “Strongly Disagree”, “Disagree”, “Neutral”, “Agree”, and “Strongly Agree”. Observed trends highlight variations in perceptions of challenges such as difficulty in surgical plane identification, increased bleeding, tissue fragility, and the quality of total mesorectal excision (TME) following TNT compared to conventional neoadjuvant therapy (CNT).

**Figure 13 cancers-17-00283-f013:**
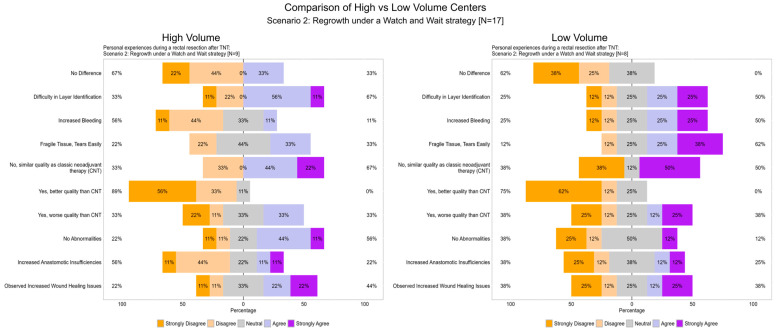
Item V.1–V.4: Subgroup analysis of personal experiences during a rectal resection after total neoadjuvant therapy (TNT) (Scenario 2). The figure compares responses from surgeons in high-volume (**left**) and low-volume (**right**) centers regarding their personal experiences during rectal resection after regrowth under a watch-and-wait strategy (Scenario 2, N = 17). Responses are presented as percentages across five Likert scale categories: “Strongly Disagree”, “Disagree”, “Neutral”, “Agree”, and “Strongly Agree”. Trends highlight variations in perceived challenges such as tissue fragility, increased bleeding, anastomotic insufficiencies, and overall surgical quality following TNT.

**Table 1 cancers-17-00283-t001:** Subgroup analysis of disagreement rates for high-volume and low-volume centers across two scenarios.

	Scenario 1			Scenario 2		
Category	High Volume	Low Volume	*p*	High Volume	Low Volume	*p*
No Difference	69% (9/13)	60% (9/15)	0.91	67% (6/9)	62% (5/8)	1
Difficulty in Plane Identification	31% (4/13)	20% (3/15)	0.41	33% (3/9)	25% (2/8)	1
Increased Bleeding	46% (6/13)	27% (4/15)	0.5	56% (5/9)	25% (2/8)	0.43
Fragile Tissue, Tears Easily	38% (5/13)	20% (3/15)	0.68	22% (2/9)	12% (1/8)	1
No, Similar Quality as CNT	31% (4/13)	47% (7/15)	0.43	33% (3/9)	38% (3/8)	1
Yes, Better Quality than CNT	77% (10/13)	60% (9/15)	0.58	89% (8/9)	75% (6/8)	0.91
Yes, Worse Quality than CNT	46% (6/13)	13% (2/15)	0.13	33% (3/9)	38% (3/8)	1
No Abnormalities	15% (2/13)	27% (4/15)	0.39	22% (2/9)	38% (3/8)	0.88
Increased Anastomotic Insufficiencies	54% (7/13)	53% (8/15)	1	56% (5/9)	38% (3/8)	0.8
Observed Increased Wound Healing Issues	31% (4/13)	33% (5/15)	0.88	22% (2/9)	38% (3/8)	0.88

The table summarizes disagreement rates (sum of “Strongly Disagree” and “Disagree”) for surgeons from high-volume and low-volume centers. Scenario 1 refers to TNT without complete clinical remission, and Scenario 2 addresses regrowth under a watch-and-wait strategy. *p*-values reflect statistical comparisons between the two groups. While no significant differences were observed (*p* > 0.05), trends highlight nuanced variations in the perception of surgical challenges and TME quality.

## Data Availability

The data presented in this study, including the full survey results, are available upon request.

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
