# Peer review of "Does Total Neoadjuvant Therapy Impact Surgical Precision in Total Mesorectal Excision? A Nationwide Survey of the Experiences of Expert Surgeons"

_cancers, 2025, doi:10.3390/cancers17020283_

Round 1
Reviewer 1 Report
Comments and Suggestions for Authors
The results of this article should be of interest to journal readers, given the current great interest in promoting the use of TNT and delayed surgery strategies.
Authors are direct with the limitations of a small survey study.
A majority of responding surgeons reported that TNT and delayed surgery strategies create tissue conditions that interfere with the precision of surgery. This is a modest but important finding deserving of consideration by stakeholders.
Reviewer 2 Report
Comments and Suggestions for Authors
The submitted paper presents the results of a nationwide (Austrian) survey of colorectal surgeons on their experience with surgery following multimodal therapy for rectal cancer. The topic is topical because of the rapid spread of "total neoadjuvant therapy" (TNT) and the lack of data on potential tissue changes that could affect surgery. The "introduction" provides an overview of the development of multimodal therapy for the treatment of locally advanced rectal cancer, not forgetting the absolute necessity of precise TME surgery. The "materials and methods" section indicates that the data presented were collected via an online questionnaire from 57 invited colorectal surgeons over a period of 4 weeks. As noted in the „results“ section, 31 responses were received, of which 30 reported experience with conventional neoadjuvant therapy (CNT) and 28 with TNT. However, the results presented are based on the highly subjective experiences and perceptions of the participants and do not represent collected patient-related data. This is a major shortcoming of the paper, but is discussed by the authors in lines 365-371.
Line62: The sentence „surgery post-TNT was commonly performed at 6-7 weeks“ is imprecise in this respect as no differentiation is made whether we are talking about induction or consolitation chemotherapy. Maybe it's the design of the questionnaire. Taken up again in:
Fig3 and Lines203-209: Don’t the authors think that induction and consolitation TNT should always be considered separately? Doesn't it make a difference whether we operate 6-7 weeks after radio(chemo)therapy or after consolitation chemotherapy?
Fig2 shows that surgeons from both high volume and low volume hospitals participated. Could you imagine that the perception and assessment of difficulties might be different?
Lines196-199: Wouldn't you agree that each type of neoadjuvant therapy leads to a very different degree of tissue change, depending on the patient?
Fig4: It is not clear to the reader why surgery after CNT started later than surgery after TNT. The basic differences between induction and consolitation chemotherapy are shown in lines 111-112 of the discussion, also the problem of different time intervals, but there is no reference to the pesented data. Aren't different waiting times the key problem of the topic?
Fig5: Neither the illustration nor the text mentions conversion rates. Wouldn’t we expect higher rates for TNT patients if we can‘t identify the layers?
Fig6: The rate of ostomy is comparable in both groups. Isn't it reasonable to assume that if we had difficulties intraoperatively, we would prefer to create a stoma?
Reviewer 3 Report
Comments and Suggestions for Authors
In this study, the authors investigated the effect of TME after neoadjuvant therapy, including TNT, on the quality of the procedure. In the conducted surveys, 56% of respondents preferred conventional neoadjuvant therapy regimens, and 32% chose TNT. Many respondents who selected TNT also reported experiencing technical challenges during TME. The authors suggest that the increased complexity and surgical difficulty following TNT may contribute to the higher local recurrence rates previously reported in TNT patients, and they advocate for further research, including refined protocols, to improve outcomes for patients with LARC.
Several points of concern are as follows:
Figure 3: It appears unusual to display the response rate for each option in a multiple-choice question as a percentage of the total responses for each choice. Is there any significance to this metric?
Questions in Items IV-2 and V-2: There is a substantial disparity between the percentage disagreeing that TME quality post-TNT is better than post-CNT and the percentage agreeing that TME quality post-TNT is worse than post-CNT. What do you believe this difference suggests?
Objectivity of the Survey: This survey relies on the subjective experience and perspectives of the respondents, lacking objective measures. Moreover, without prognostic studies included, attributing the higher local recurrence rate after TNT to increased surgical difficulty lacks sufficient persuasive evidence.
